# Packaging contests between viral RNA molecules and kinetic selectivity

**Inbal Mizrahi[1], Robijn Bruinsma[1,2]\*, Joseph Rudnick[1]**

**1** Department of Physics and Astronomy, University of California, Los Angeles, California, United States of America, **2** Department of Chemistry and Biochemistry, University of California, Los Angeles, California, United States of America

\* bruinsma@physics.ucla.edu

## Abstract

The paper presents a statistical-mechanics model for the kinetic selection of viral RNA molecules by packaging signals during the nucleation stage of the assembly of small RNA viruses. The effects of the RNA secondary structure and folding geometry of the packaging signals on the assembly activation energy barrier are encoded by a pair of characteristics: the *wrapping number* and the *maximum ladder distance*. Kinetic selection is found to be optimal when assembly takes place under conditions of *supersaturation* and also when the concentration ratio of capsid protein and viral RNA concentrations equals the stoichiometric ratio of assembled viral particles. As a function of the height of the activation energy barrier, there is a form of order-disorder transition such that for sufficiently low activation energy barriers, kinetic selectivity is erased by entropic effects associated with the number of assembly pathways.

## Author summary

During the assembly of a viral particle, a limited number of viral genomic RNA molecules must compete for packaging with a large number of closely similar host messenger RNA molecules. All-atom simulations of this competition process are impractical. The paper presents a tractable mathematical model for the selection process as a *non-equilibrium* phenomenon.

## Introduction

When the molecular components of a single-stranded (ss) RNA virus assemble and form virions in the cytoplasm of an infected cell, genomic viral RNA molecules (gRNA) compete for packaging with a large pool of—quite similar—host messenger RNA (mRNA) molecules for packaging by the viral capsid proteins [1]. For example, for the case of influenza the number of gRNA molecules inside an infected cell is less than $10^4$ [2] while the total number of host mRNA molecules is in the range of $3.6 \times 10^5$.

**Data Availability Statement:** All relevant data are within the manuscript and its Supporting information files.

**Funding:** This study was funded by National Science Foundation, Directorate for Mathematical

and Physical Sciences, CMMT Grant No.1836404. https://www.nsf.gov/funding/pgm_summ.jsp? pims_id=505357 (to R.B.). The funders had no role in study design, data collection and analysis, decision to publish, or preparation of the manuscript.

**Competing interests:** The authors have declared that no competing interests exist.

Viral RNA selection relies on so-called *packaging signals* [3–9]. These are short RNA stem loop motifs that are a part of the secondary structure of gRNA molecules. Importantly, *any* ssRNA molecule with the molecular weight of a gRNA molecule has numerous hairpins, some of which may be similar or identical to one of the packaging signals of a virus. In order to avoid the packaging of mRNA material *individual* packaging signals should not trigger virion assembly. Viral gRNA molecules typically have a coordinated pattern of packaging signals that collectively direct the assembly, sometimes called a "$\psi$ sequence". These virus-specific interactions between packaging signals and capsid proteins operate together with a generic, non-specific electrostatic affinity between the negatively charged RNA nucleotides and positively charged residues of the capsid proteins [10–13].

It has been demonstrated that the spontaneous self-assembly of empty icosahedral capsids is initiated by the formation of a *nucleation complex* composed of a limited number of capsid proteins [14–16]. This nucleation complex can be compared to the *critical nucleus* of the kinetic theory of nucleation and growth [17, 18]. The energetically uphill formation of the nucleation complex is followed by the energetically down-hill growth (or "elongation") process that ends in the formation of a closed capsid. The self-assembly of empty capsids does not (and should not) take place under physiological conditions. Under those conditions electrostatic repulsion between the capsid proteins is just able to overcome the hydrophobic affinity between capsid proteins. The physical aspects of RNA packaging have been extensively studied experimentally and theoretically [10, 11, 19–35] as well as by numerical modeling [13, 36–38]. For reviews see refs. [18, 39, 40]. Theoretical models have generally focused on the minimization of the free energy of a fully assembled viral particle. This produced global measures for the "packaging fitness" of an RNA molecule in terms of its length, the degree of branching and compactness, and the effects of electrostatics and osmotic pressure.

The pioneering work by Aaron Klug on TMV [41] showed that the action of gRNA on assembly is two-fold. On the one hand, negative charges of the RNA molecules neutralize—on a non-specific basis—positive capsid protein charges. This shifts the overall equilibrium free-energy balance from a dispersed state towards aggregation. On the other hand, the specific packaging signals on gRNA act as *catalysts* that lower the activation energy barrier of the nucleation complex. In the view of Klug, the packaging signals affect the assembly *kinetics* while in a thermodynamic view the role of the packaging signals would be to further tilt the free energy balance in favor of packaging. The most well-studied case of RNA selectivity is probably that of the HIV-1 retrovirus (see ref. [42] and references therein). RNA selectivity depends on the cooperative action of a cluster of packaging signals located at the 5' end of the gRNA molecule, the $\psi$ sequence. It is about a hundred nucleotides long, which is very small compared to the total length of the HIV-1 genome (about $10^4$ nucleotides). gRNA selection appears to take place very early, during the nucleation stage of the assembly process when the $\psi$ sequence interacts with only a small group of capsid proteins. Changing the RNA sequence of other sections of the genome molecules does not appear to affect the selectivity. *The gRNA molecules appear to have no thermodynamic advantage over non-viral RNA molecules of the same length* [43]. This indicates that the origin of the very efficient gRNA selection mechanism of HIV-1 must be sought in the kinetics of the assembly process. Finally, recent progress in the asymmetric image reconstruction of certain small RNA viruses [44] indicate that also in those cases the RNA selection process takes place early in the assembly process. Asymmetric reconstruction of the MS2 phage virus [45] shows that RNA packaging signals associate *reproducibly* with a specific section of the interior of the capsid. The authors proposed a model for viral assembly in which a spatial distribution of packaging signals functions as a virus-specific "map" for the initial nucleation stage of the assembly while the subsequent elongation step of the assembly is driven more by non-specific interactions. This scenario appears similar to that

of the TMV and HIV-1 viruses. There certainly are also counter examples where free energy minimization accounts for selection. For example, the asymmetric reconstruction of the CCMV and BMV plant viruses produced only a very small amount of reproducible RNA-protein association [46]. Interestingly, this group of viruses is much less selective than MS2. CCMV capsid proteins appear to select BMV genome molecules over CCMV genome molecules while they can package a wide variety of non-viral ss RNA molecules and even non-RNA polyelectrolytes [47, 48]. Apparently, the amount of CCMV gRNA molecules produced inside an infected cell is sufficiently large so there is no need for very precise assembly selectivity.

In this paper we propose a simple statistical-physics model to study the physics of selective nucleation by a group of packaging signals that encode the assembly of a small ssRNA virus. By construction, the model focuses exclusively on kinetic selection. In the conclusion we will return to the relation between the thermodynamic and kinetic modes of selection.

## Model and methods

### Spanning tree model

The starting state of the system is assumed to be a solution containing a certain concentration of condensed, folded viral RNA molecules and of pentameric capsid proteins. The folded molecules have the same interior structure and dimensions and differ only in terms of the $\psi$ section of contiguous packaging signals distributed over the surface of the condensed RNA molecule. The capsid of the virus is assumed to be composed of twelve protein pentamers assembled into a dodecahedron such that double-stranded (ds) RNA sequences line the edges of the pentamers. This is inspired by the family of the *Nodaviridae* in which part of the ssRNA genome are ds sequences forming a dodecahedral cage [49]. The secondary structure of the $\psi$ section is represented as a tree of nineteen links and twenty nodes connecting the vertices of the dodecahedron. The pentameric capsid itself is the well-studied Zlotnick model system for empty capsids [50–53]). The geometry of the $\psi$ section is assumed to be adapted to the dodecahedral capsid so that the twenty nodes of the secondary structure match up with the twenty vertices of the dodecahedron. Despite these constraints there still are tens of thousands of secondary structures that satisfy these constraints. In the mathematical literature, these structures are known as the *spanning tree graphs* of a dodecahedron [54]. The number of nodes of a spanning tree is twenty because a spanning tree must visit all the vertices, of which there are twenty. The number of links is nineteen because in any connected tree graph the number of links is one less than the number of nodes. A spanning tree leaves eleven edges of the dodecahedron uncovered. We will assume that these remaining edges have only a generic affinity for the capsid proteins. Fig 1 shows an example of a spanning tree of the dodecahedron.

Initially, all pentamers are in solution at a certain total concentration $c_0$. The pentamers are assumed to have a generic affinity (electrostatic in actuality) for all edges of the dodecahedron plus a specific affinity for those edges that are covered by links of the spanning tree (acting as packaging signals). The specific affinity is maximized by placing the pentamer on a location such that four of its edges can associate with a link of the chain. For the tree molecule shown in Fig 1, a total of six pentamers can be placed on such maximum wrapping locations. We will say that the *wrapping number* of this tree structure is $N_P = 6$. The wrapping number is a characteristic of the folding geometry of the RNA molecule.

The very simplest spanning tree is a linear chain composed of nineteen links. Fig 2 shows how a linear chain can be distributed over a dodcahedron, while visiting all vertices. As shown in Fig 2, only two pentamers can be placed on locations such that four of its edges associate with a link of the chain. There are 1620 different configurations for a linear chain of twenty nodes to be distributed over of a dodecahedron such that the nodes coincide with the vertices

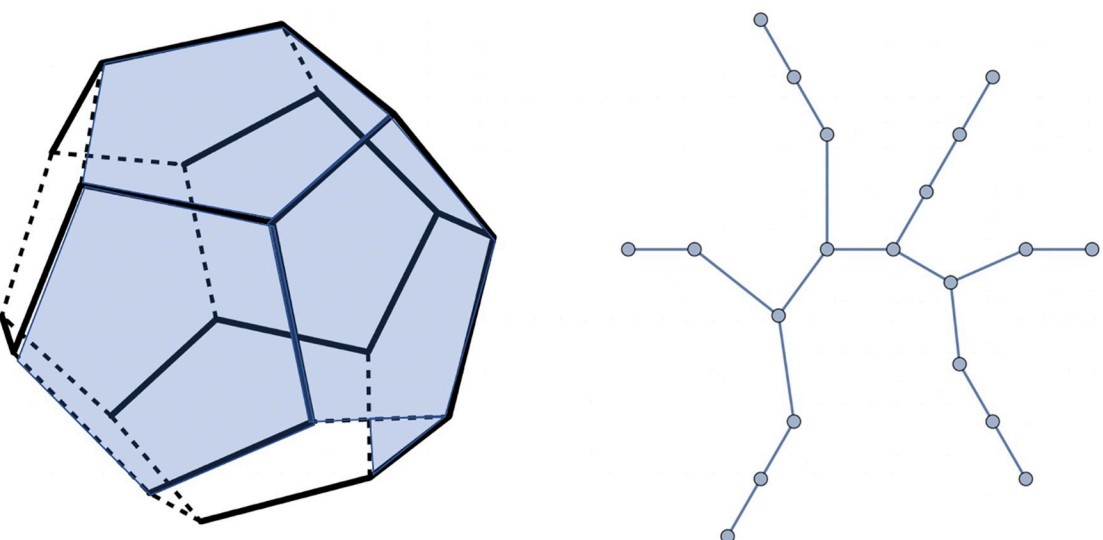

**Fig 1.** Left: Branched spanning tree connecting the vertices of a dodecahedron. Six pentamers can be placed on the dodecahedron such that their edges make the maximum of four contacts with links of the spanning tree. Right: Planar representation of the spanning tree.

known in the mathematical literature as *Hamiltonian paths*. Hamiltonian paths have been used before to classify viral RNA configurations [8, 55, 56]). Hamiltonian paths of the dodecahedron can have wrapping numbers two, three, or four. There are important differences between the linear and the branched cases. In the linear case, pentamers have embeddings with two, three or four edges occupied while for the branched case, there are embeddings with zero, one, two, three or four edges occupied.

Next, the edges of a pentamer will be assumed to have affinity for the edges of other pentamers. It is this affinity that drives the assembly of empty capsids. The wrapping number does

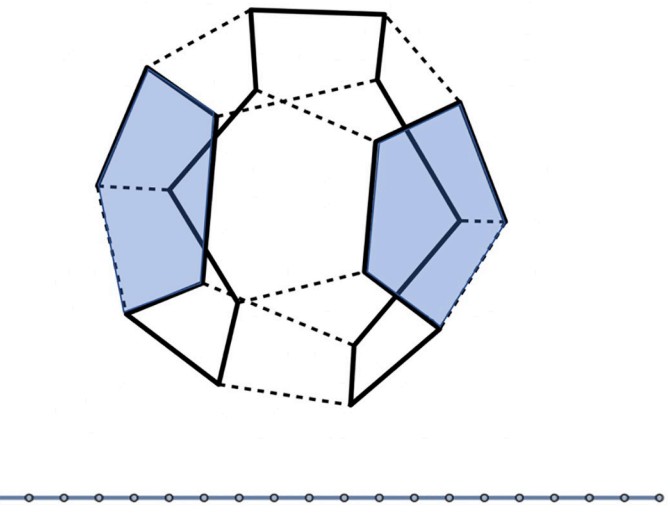

**Fig 2. Example of a spanning tree in the form of a Hamiltonian path of a dodecahedron.** In blue are shown two pentamers that can be placed on the dodecahedron such that their edges make the maximum of four contacts with links of the spanning tree.

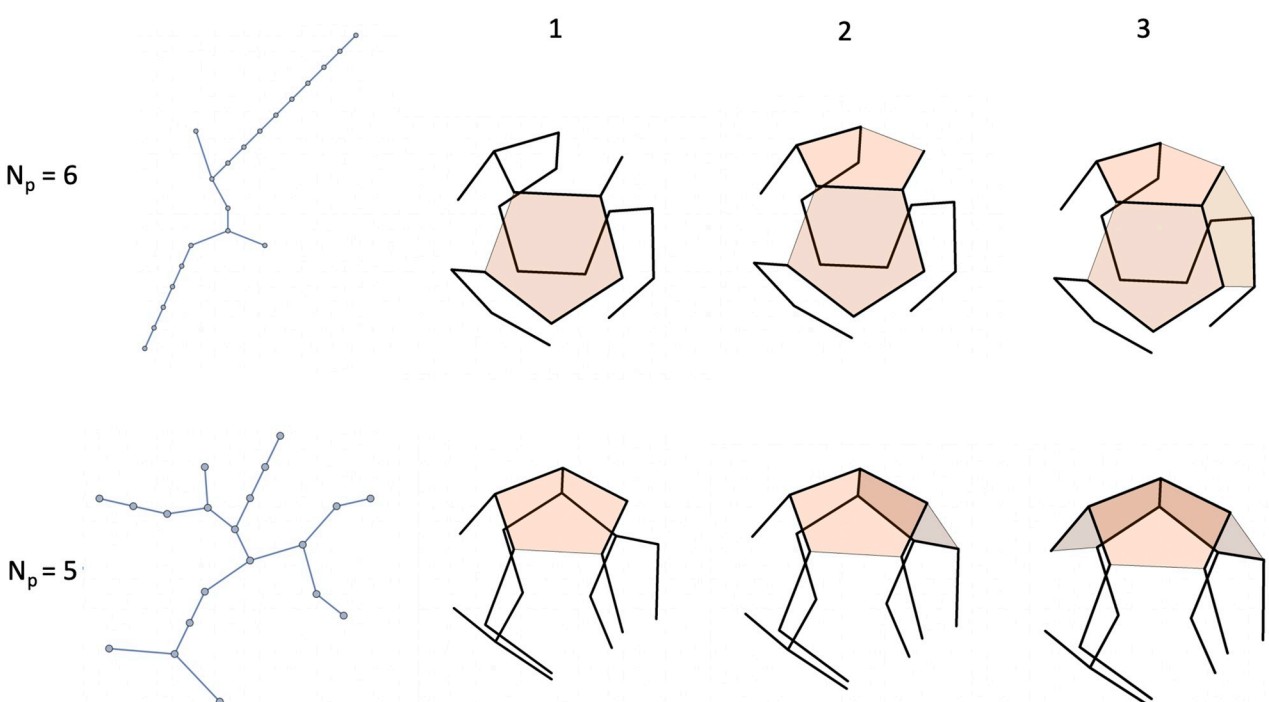

**Fig 3. The first three steps of the minimum energy assembly pathways of two molecules, with wrapping numbers $N_p = 6$ (top) and $N_p = 5$ (bottom) respectively, for the case that the affinity of pentamer edges for each other exceeds the specific affinity for the spanning tree.** The more compact $N_p = 5$ tree allows three pentamers on sites with four links with each pentamer making two edge-to-edge contacts. For the less compact $N_p = 6$ tree, the third pentamer only makes three contacts with a link.

not measure how many attractive contacts a newly added pentamer can make with pentamers that were placed earlier on the dodecahedron. The more *compact* a spanning tree, the larger the probability that two maximally wrapped pentamers also are able to share an edge. Fig 3 illustrates the important role of compactness of the spanning tree. The figure shows the first three steps of the minimum energy assembly pathways of two spanning trees, one with a wrapping numbers $N_p = 5$ and the other with $N_p = 6$. It is assumed that the edge-edge affinity exceeds the edge-link affinity. In both cases, the first two pentamers can be placed on adjacent sites with maximum wrapping so they have one shared edge. The assembly energy is the same at this point. A difference appears when the third pentamer is placed on the assembly. For the more compact $N_p = 5$ molecule, the third pentamer can be placed on a maximum wrapping site where it has two shared edges with the two pentamers already present but this is not possible for the less compact $N_p = 6$ tree. It follows that the minimum assembly energy of a three pentamer cluster for the $N_p = 5$ molecule is lower than that of the $N_p = 6$ molecule. The wrapping number is thus, by itself, insufficient as an index that can predict which spanning trees favor assembly nucleation.

The Maximum Ladder Distance (MLD) has been used to characterize the degree of compactness of the secondary structure of complete gRNA molecules and as a measure of the size of an RNA molecule in solution [57, 58]. In the mathematical literature, the ladder distance (or LD) between two nodes of a tree graph is defined as the number of links of the graph along a minimum length path separating the two nodes. The MLD of a tree graph is the largest LD of the graph. In the language of graph theory, the LD between two nodes of a tree graph is known as the "distance" between the two nodes and the MLD as the "diameter" of a tree graph [59].

Below we apply the MLD concept only to the $\psi$ section of the complete molecules, not to the RNA molecule as a whole. The MLD of the branched spanning tree that was just discussed is nine while it is nineteen for the linear tree. Unlike the wrapping number, the MLD is a characteristic that is determined entirely by the topology of the planar graph of the secondary structure of the $\psi$ sequence. Unlike the wrapping number, it remains the same for different folding geometries. We will see that the combination of the wrapping and MLD numbers together forms a satisfactory index for the effectiveness of a spanning tree as a nucleation catalyst.

## Minimum-energy assemblies

A spanning tree/pentamer structure with $n$ pentamers is assigned an assembly energy

$$\Delta E(n) = E_0(n_1\epsilon_1 + n_2\epsilon_2 - n_3 - \mu_0 n)$$

with respect to an RNA molecule without pentamers. Here, $n_1$ is the number of links of the spanning tree that lie along a pentamer edge that is not shared with another pentamer, $n_2$ is the number of spanning tree links that lie along a pentamer edge that is shared with another pentamer while $n_3$ is the number of edges shared between two pentamers that are not associated with a link of the spanning tree. The energy scale $E_0$ is the binding energy between two pentamer edges in the absence of specific RNA-pentamer affinity. Energies will be expressed in units of $E_0$. The physical meaning of the dimensionless parameter $-\epsilon_1$ is that of the ratio of the affinity of a spanning tree link for a pentamer edge over $E_0$. Interactions between edges and spanning tree links will be assumed to be additive. In that case the dimensionless $\epsilon_2$ parameter equals $\epsilon_2 = -1 + 2\epsilon_1$ since the RNA link interacts with two pentamer edges. Finally, $\mu_0$ is the chemical potential of pentamers in solution at a certain reference concentration. The reference chemical potential includes the non-specific affinity of a pentamer for the RNA condensate. The reference chemical potential will be chosen so that $\Delta E(12)$ is close to zero, so when the chemical potential in solution at the reference concentration is the same as that of a pentamer that is part of an assembled capsid. This is the case if $\Delta E(12) = 19\epsilon_2 - 11 - 12\mu_0$ is zero. Note that $\Delta E(12)$ is the same for all spanning trees so different spanning trees have the same assembly energy (we will see later that the assembly *free* energy is not the same for all trees).

Two examples of minimum-energy assembly profiles near assembly equilibrium with $c_0 = 1$ are shown in Fig 4 The left figure shows the case of an $N_P = 8$, $MLD = 9$ spanning tree. Recall that such a spanning tree is maximally adjusted for pentamer binding. The right figure shows the case of an $N_P = 2$, $MLD = 19$ spanning tree, which has minimal adjustment for pentamers. The activation energy is about two $E_0$ higher in the second case. The assembly energy profiles of spanning trees with the same $N_P$ and $MLD$ are nearly always the same.

These assembly energy profiles are consistent with a nucleation-and-growth scenario close to the equilibrium assembly threshold. As expected, the activation energy barrier of the $N_P = 8$, $MLD = 9$ spanning tree (about $3.0E_0$) is lower than that of the $N_P = 2$, $MLD = 19$ spanning tree (about $5.0E_0$). The long straight section of the $N_P = 8$, $MLD = 9$ energy profile can be understood by noting that when a pentamer is added to one of the eight maximum wrapping sites then that lowers the energy by $4\epsilon_1$ in units of $E_0$. If additional pentamers always make two new contacts with pentamers that are already present—as is indeed the case here—then each added pentamer lowers the energy further by an amount of $2 E_0$ minus the chemical potential $\mu_0$. In this particular case, these two terms cancel so the assembly energy barrier has a wide and flat top. Starting from a linear chain with MLD = 19 and $N_p = 2$ and then stepwise decreasing the MLD and increasing the $N_p$ one finds that the assembly energy activation barrier nearly always systematically decreases. Assembly on a spanning tree with the minimum MLD and the

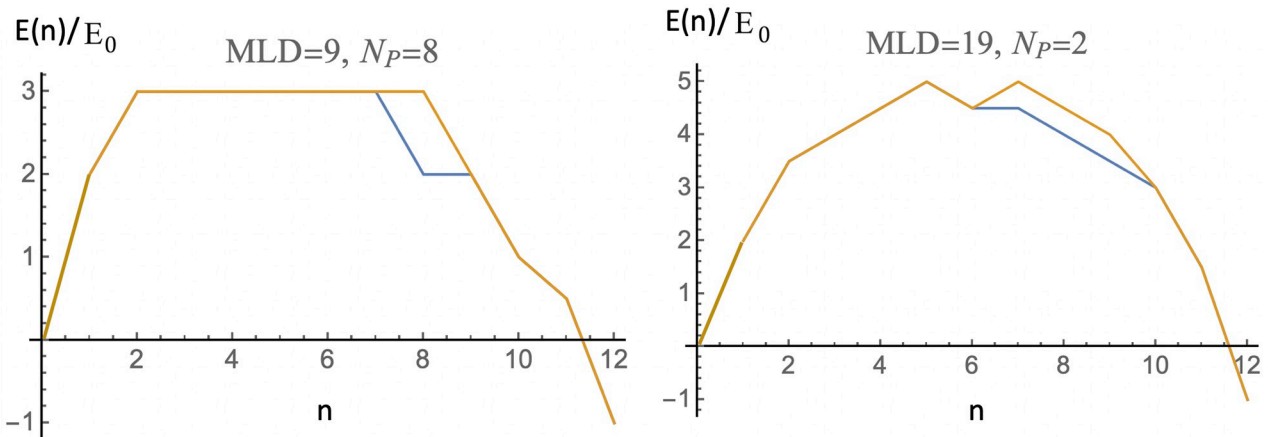

**Fig 4. Minimum-energy assembly energy profiles for $N_P = 8$, $MLD = 9$ spanning trees (left) and for an $N_P = 2$, $MLD = 19$ spanning tree (right).** Energy parameters are $\epsilon_1 = -0.5$ and $\mu_0 = -4.0$ (close to assembly equilibrium where $\mu_0 \simeq -4.083$). Energies are expressed in units of the overall scale $E_0$. The assembly energy profiles of spanning trees with the same $N_P$ and $MLD$ are nearly always the same. The pathways for the small number of exceptions are shown in the figure.

maximal wrapping number $N_p$ has in general the lowest possible assembly activation energy barrier.

An illustration of the assembly sequence of an $N_P = 8$, $MLD = 9$ spanning tree spanning tree is shown in Fig 5 for the case that $-\epsilon_1$ is less than 0.5. The first eight pentamers all can be placed on sites that maximize the number of spanning tree link contacts (four) as well as the number of attractive pentamer-pentamer contacts. The second pentamer creates one new pentamer-pentamer contact while the next three pentamers create two new pentamer-pentamer contacts. All pentamer assembly intermediates are compact structures. In summary, the combination of the wrapping number and the MLD appears to be a good, though not perfect, code for the height of the assembly energy activation barrier.

Next, we explored the configuration space of assembly intermediates. Fig 6 shows graphs of all distinct assembly intermediates for molecule (1) and (2) as well as the minimum energy assembly pathways that link them. Each node of the network stands here for a physically distinct assembly intermediate (assemblies that are related by a symmetry operation of the dodecahedron are treated here as the same). Nodes are assigned "coordinates" $(n, i)$ with $n = 0$, 1, . . .., 12 the number of pentamers of the intermediate and with $i = 1, 2, . . . . . ., m_n$ an index ranging over the distinct n-pentamer states where $m_n$ is the *multiplicity* of the n-pentamer state (e.g., $m_5 = 4$ for the $N_P = 8$, $MLD = 9$ spanning tree). A black line linking two dots indicates that the two states can be interconverted by addition or removal of a pentamer. Assembly of viral particles can be viewed as a net "current" flow from the $n = 0$ source state to the $n = 12$ final state along all possible paths across the network linking the initial state to the final state. Under conditions of thermodynamic equilibrium, the current across each individual link should be zero according to the principle of detailed balance. Note that the compact, branched spanning tree molecule (1) (left) has far fewer assembly intermediates and assembly pathways than the linear structure molecule (2) (right).

An important simplification ensues when we ignore the very small number of assembly energy profiles that do not conform to the quasi-universal profile for given $N_P$ and $MLD$ discussed above. If we do that then *different nodes of the network with the same n all have have the same assembly energy $\Delta E(n)$*. This simplification allows us assign to each node a Boltzmann

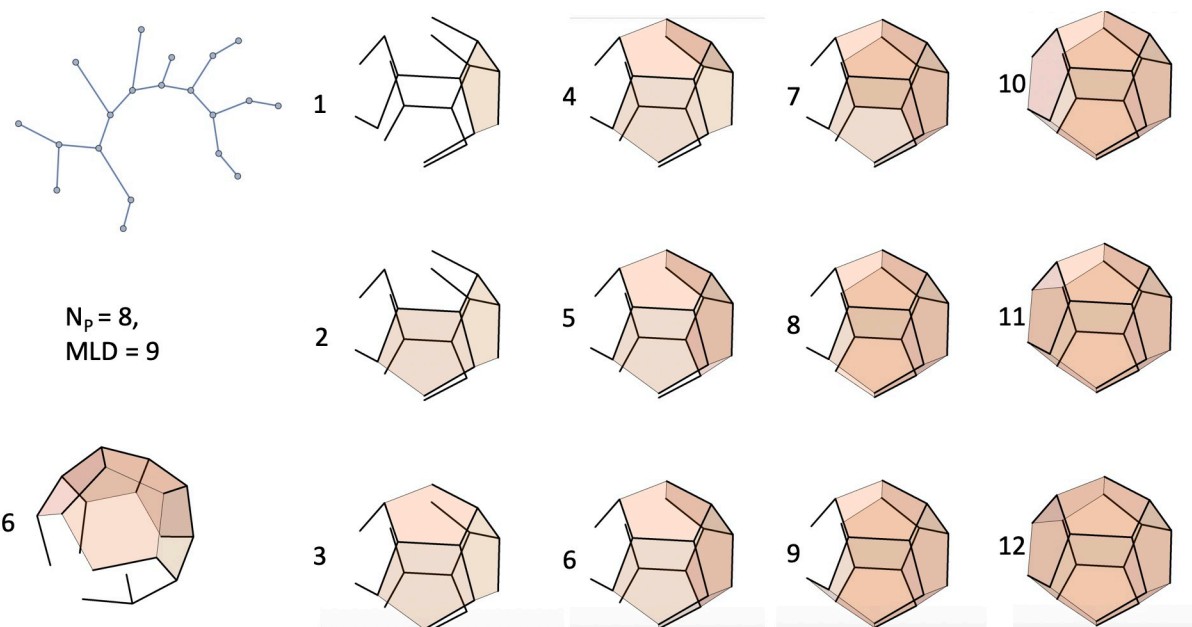

**Fig 5. A spanning tree with a maximum wrapping number of eight and the minimum MLD of nine.** The first five pentamers can be placed on sites that maximize both the number of available pentamer-pentamer contacts and pentamer-spanning tree link contacts (four). The sixth pentamer, shown separately with a different perspective, makes three pentamer-pentamer contacts but can only make two spanning tree link contacts.

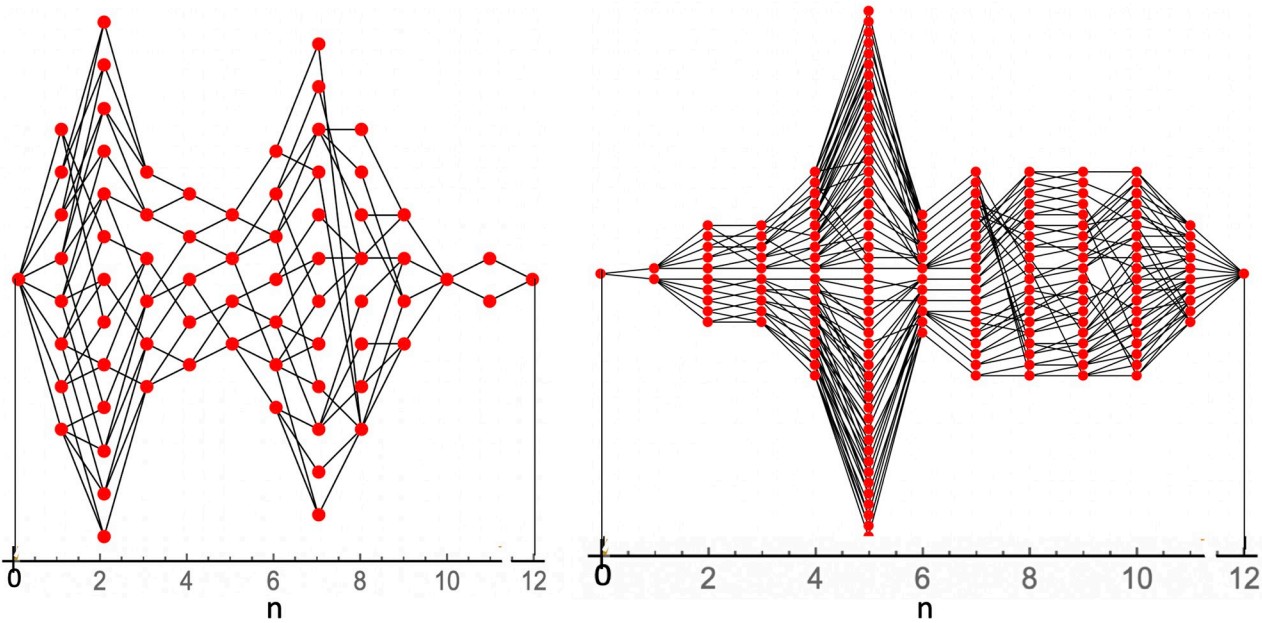

**Fig 6. Examples of minimum energy assembly paths of an $N_P = 8$, $MLD = 9$ spanning tree (left) and an $N_P = 2$, $MLD = 19$ spanning tree (right).** A node (indicated by a dot) indicates a physically distinct intermediate structure with, from left to right, $n = 0, 1, \ldots, 12$ pentamers. The number $m_n$ of vertical dots for given $n$ is the *multiplicity*, i.e., the number of distinct n-pentamer intermediates. A link connecting two nodes indicates that the two states are related by addition or removal of a pentamer. Every possible path from n = 0 to n = 12, including back steps, represents a possible minimum energy assembly pathway. During assembly there is a net current from the n = 0 initial state to the n = 12 final state while in thermal equilibrium the net current across every link is zero.

equilibrium probability:

$$P_{eq}(n) \propto \exp - \beta \Delta E(n) + n \, \ln(c_f) \tag{1}$$

which depends on the concentration $c_f$ of pentamers free in solution (expressed in units of the reference concentration). The proportionality factor in the expression is determined by the normalization condition $\sum_{n=1}^{12} m_n P_{eq}(n) = 1$. We will make this simplification in the following sections.

## Master equation

In this section we define the Master Equation that governs the kinetics. We will use the coordinate system for the network graphs defined below Fig 6. The network geometry of a particular spanning tree is specified in the form of an *adjacency matrix* $A_n^{i,j}$ that equals one if a link connects node $n$, $i$ to node $n + 1$, $j$ while it equals zero if there is no link. Each node $n$, $i$ of the graphs has a time-dependent occupation probability $P_{i,n}(t)$. The kinetics is expressed as a set of coupled rate equations for the $P_n^i(t)$ and is assumed to be a Markov process with probabilities evolving in time according to the Master Equation [60]:

$$\frac{dP_{i,n}(t)}{dt} = \sum_j \{A_{n-1}^{j,i} W_{n-1,n} P_{j,n-1}(t) + A_n^{i,j} W_{n+1,n} P_{j,n+1}(t)\}$$
$$- P_{i,n}(t) \sum_j \{A_{n-1}^{j,i} W_{n,n-1} + A_n^{i,j} W_{n,n+1}\} \tag{2}$$

Here, $W_{n,n+1}$ is the on-rate for the transition of an assembly of $n$ pentamers to one with size $n + 1$ by the addition of a pentamer while $W_{n,n-1}$ is the off-rate at which a pentamer is removed from an assembly of size $n$. We assume a simplified diffusion-limited chemical kinetics (see [61] and Supplementary Information S1 Text) in which the addition or removal of a pentamer to an assembly of size $n$ is treated as a bimolecular reaction. The resulting on-rate has the form of a kinetic Monte-Carlo algorithm:

$$W_{n,n+1} = \lambda c_f \begin{cases} e^{-\Delta\Delta E_{n,n+1}} & \text{if} \quad \Delta E(n+1) > \Delta E(n) \\ 1 & \text{if} \quad \Delta E(n+1) < \Delta E(n) \end{cases} \tag{3}$$

with $c_f$ again the concentration of free pentamers, $\Delta\Delta E_{n,n+1} = \Delta E(n + 1) - \Delta E(n)$ the energy cost of adding a pentamer, and $\lambda$ a base rate that depends on molecular quantities such as diffusion coefficients and reaction radii but not on the concentrations. The inverse of $\lambda$ is the fundamental time-scale of the kinetics. In the following, time will be expressed in units of $1/\lambda$. If $\Delta\Delta E_{n,n+1}$ is negative then the on-rate is equal to this base rate. If $\Delta\Delta E_{n,n+1}$ is positive then the base rate is reduced by the Arrhenius factor $e^{-\Delta\Delta E_{n,n+1}}$. The off-rate entries $W_{n+1,n}$ are determined by the on-rates through the condition of detailed balance:

$$\frac{W_{n,n+1}}{W_{n+1,n}} = \frac{P_{eq}(n+1)}{P_{eq}(n)} = c_f e^{\Delta\Delta E_{n,n+1}} \tag{4}$$

Below, we will limit ourselves to the case of solutions containing only two species of spanning trees, namely molecules (1) and (2), having the same solution concentrations. During assembly, the two species compete for the same concentration $c_f$ of pentamers. The ratio of $c_f$

over the total pentamer concentration $c_0$ is determined by mass conservation:

$$c_f/c_0 = 1 - (D/24)\sum_{n=0}^{12}\left(\sum_{i=1}^{m_n^{(1)}}nP_{i,n}^{(1)}(t) + \sum_{i=1}^{m_n^{(2)}}nP_{i,n}^{(2)}(t)\right) \tag{5}$$

where superscripts denote the kind of spanning tree. Next, $D \equiv 12r_t/c_0$, with $r_t$ the total RNA concentration, is the RNA to protein *mixing ratio*, an important thermodynamic control parameter for the assembly process. If $D = 1$ then there are exactly enough pentamers to encapsidate both types spinning trees, which corresponds to the *stoichiometric ratio*. Since the occupation probabilities that we seek to obtain enter in this relation, the rate equations form a coupled, non-linear set of two times twelve differential equations (for the case of just a single species in solution, there are twelve differential equations where one must replace $D/24$ by $D/12$).

## Numerical construction and implementation of coupled master equations with competition

Coupled master equations for packaging competition between pairs of spanning trees were integrated using a Mathematica program (available on request). The program was organized as follows.

**First step: Construction of spanning trees.**   First, we populate in all possible ways nineteen of the thirty one edges of the dodecahedron with links. Next we identify graphs that have the properties that (i) the graph is connected and that (ii) the graph has twenty vertices. All graphs having these two properties are spanning trees because (i) any connected graph with $n$ edges and $n + 1$ vertices is a tree and (ii) they are spanning trees because a dodecahedron has twenty vertices. This method generates many duplicates so the next step is winnowing the trees down to a collection of trees that cannot be mapped into each other by any of the 120 rotations and reflections that map a dodecahedron into itself. This is a straightforward (though laborious) process that involves choosing a tree and then eliminating all other trees that can be mapped into it by a reflection or a rotation. One can speed the process up by separating trees into subsets having the same MLD, as the MLD is a topological property that is preserved under rotation and reflection.

**Second step: Choice of a pair of spanning trees.**   Two trees are chosen from the library of all unique spanning trees of the dodecahedron. The trees are indexed by their MLD and $N_P$ numbers.

**Third step: Specification of the assembly energies.**   Values are assigned to the energy parameters $E_0$, $\epsilon_1$ and $\mu_0$, as defined above.

**Fourth: Determination of the assembly network.**   All physically distinct minimum energy assemblies are determined forming the nodes of the assembly graph. For every node, a list is made of nodes with one more or or one less pentamer from which the adjacency matrix is determined.

**Fifth step: Construction of the master equation.**   The assembly/disassembly process is assumed to occur by the addition of a pentamer to, or removal of a pentamer from, a structure. The steps of addition or removal are controlled by two considerations. The first is the increase or decrease in the energy of the system as determined by the energy of the partially or completely assembled capsid and an assigned chemical potential. The second is the availability of pentamers, quantified by the concentration in solution of available pentamers. The free energy increment determines the relative likelihoods of addition or removal of a pentamer, via the kinetic Monte-Carlo rates that were defined above. The concentration of available

pentamers controls the rate of the assembly/disassembly process, in that disassembly takes place at a fixed rate and assembly at a rate proportional to the number of available pentamers.

## Time-scales

As a preliminary, Figs 7 and 8 show the packaging kinetics of a single species of RNA molecules. The parameters are $E_0 = 4k_bT$, $\epsilon_1 = -0.5$, $c_0 = 1$ and $D = 0.5$ and $\mu_0 = -4$. Fig 7 shows the case of the class of MLD = 9 and $N_p = 8$ spanning trees and Fig 8 that of the class of MLD = 19, $N_p = 2$ spanning trees. In the late time limit, the two classes approach thermal equilibrium with roughly the same fraction of RNA molecules being packaged (about sixty six percent). This reflects the fact that all classes of molecules have—by construction—the same assembly energy (the remaining small difference is due to the fact the entropy in the assembly free energy is not the same for the two classes). The reason that in both cases a significant fraction of RNA molecules has not been packaged, despite the fact that there are twice as many pentamers as needed to package the RNA molecules, is that the chemical potential is close to assembly equilibrium so a significant number of RNA molecules remain free of pentamers. There is a large difference between the thermalization times of two classes, roughly $10^3$ time units for MLD = 9 and $N_p = 8$ spanning trees and $10^5$ time units for MLD = 19, $N_p = 2$ spanning trees, consistent with the fact that the assembly activation barrier is about two times $E_0$ larger (i.e., about $8k_bT$) for the MLD = 19, $N_p = 2$ spanning trees. These thermalization times must be compared with the assembly *delay time* $t_d$, which is defined as the time lag between the establishment of solution assembly conditions and the first appearance of assembled viral particles. Measured delay times for the assembly of empty capsids are in the range of seconds to minutes [14–16]. We obtain $t_d$ from the intersection of the tangent to $P_{12}(t)$ at the point of maximum slope with the horizontal axis (see Fig 9).

For the case of the MLD = 9 and $N_p = 8$ class of spanning trees, this gives about 8.5 time units as shown. Other classes have comparable delay times. It follows that our time unit is roughly in the range of five seconds. The thermalization time under conditions of assembly equilibrium would then be in the range of a prohibitively long *two hundred hours* for MLD = 9, $N_p = 8$ spanning trees and two orders of magnitude longer for the MLD = 9, $N_p = 8$ spanning trees.

## Packaging competition

In Figs 10, 11 and 12 we show the result of a calculation with the same total amount of RNA molecules and pentamers as before but now with half of the RNA molecules MLD = 9 and $N_p = 8$ spanning trees and the other half MLD = 19, $N_p = 2$ spanning trees.

The MLD = 9, $N_p = 8$ spanning trees dominate packaging on time scales less than about $10^7$ time units while the characteristic assembly time is in the range of $10^4$ time units. About eighty percent of these spanning trees are packaged. This packaged fraction then slowly decreases on time scales of the order of $10^7 - 10^8$ time unit, which means that packaged MLD = 9, $N_p = 8$ spanning trees are gradually *disassembling* as a precursor of thermalization. The fraction of packaged MLD = 19, $N_p = 2$ spanning trees increases correspondingly. Apparently, pentamers that are being freed up by disassembly of MLD = 9, $N_p = 8$ spanning trees feed assembly of the MLD = 19, $N_p = 2$ spanning trees. When the system approaches thermal equilibrium, the packaging fractions of the two classes in the long-time limit are nearly the same and close to the separate equilibrium values found earlier in the absence of competition. We conclude that in a packaging competition experiment, the disassembly of particles is a crucial step for reaching thermodynamic equilibrium.

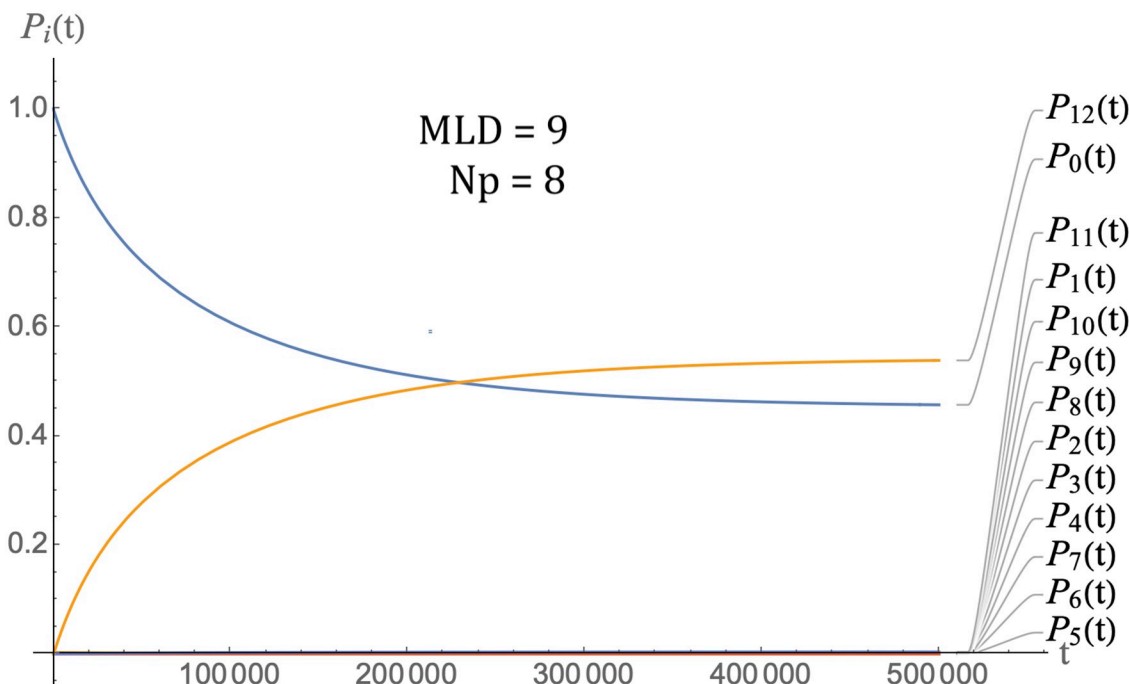

**Fig 7. Packaging kinetics of MLD = 9 and $N_p$ = 8 spanning trees.** Parameter values are $E_0 = 4k_bT$ for the energy scale, $\epsilon_1 = -0.5$, $c_0 = 1$, $D = 0.5$ and $\mu_0 = -4$. Bottom: Packaging kinetics of MLD = 19, $N_p$ = 2 spanning trees with the same parameters.

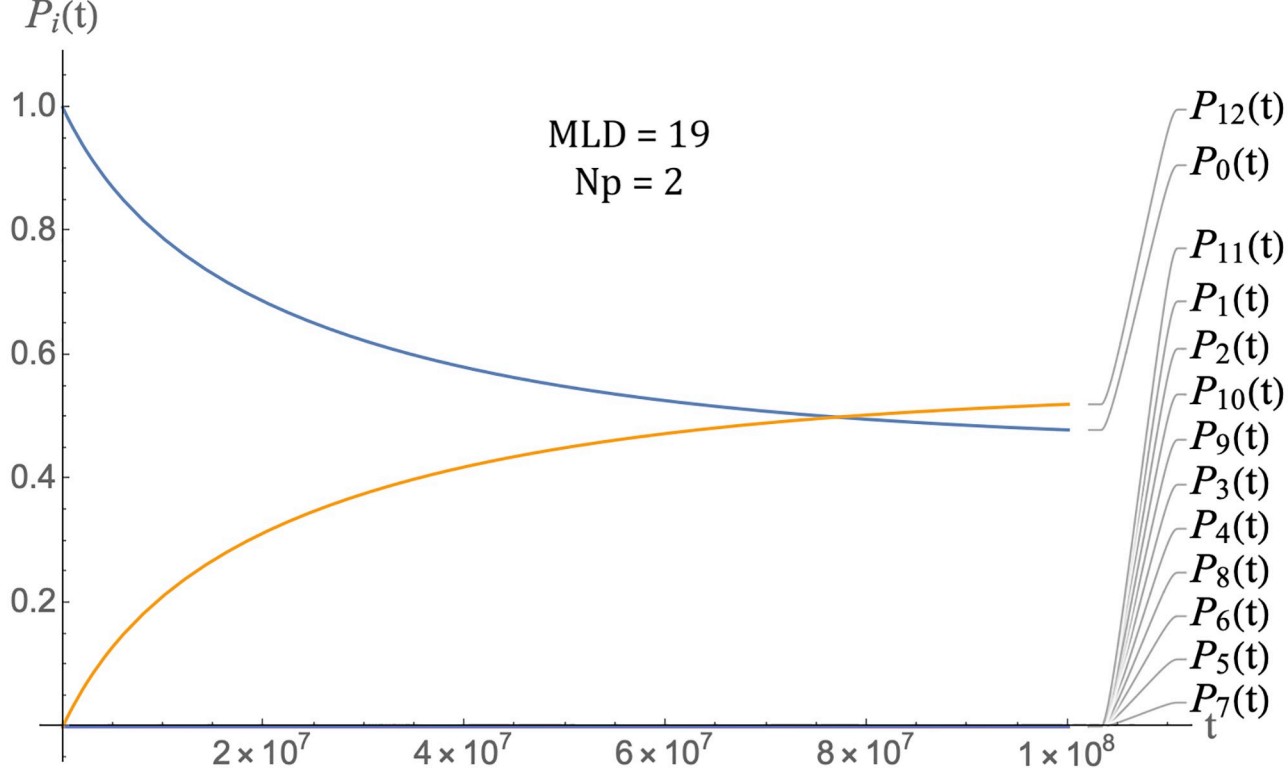

**Fig 8. Packaging kinetics of MLD = 19, $N_p$ = 2 spanning trees with the same parameters as those of the previous figure.**

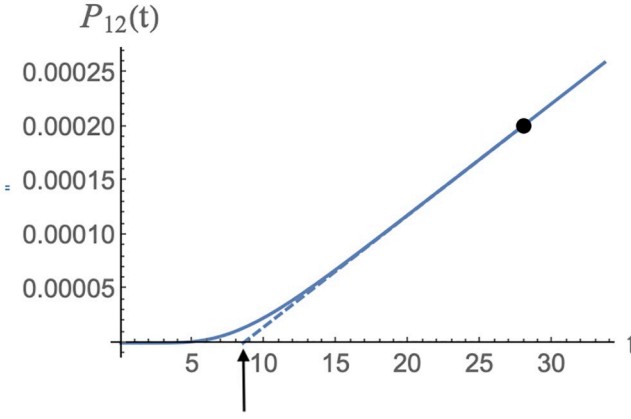

**Fig 9. Definition of the delay time as the intersection of the maximum slope tangent to the assembly curve $P_{12}(t)$ with the time axis, as indicated by the arrow.** $MLD = 9$, $N_p = 8$, $\epsilon = 0.5$, $c_0 = 1$, $D = 0.5$ and $\mu_0 = -4$.

## Order-disorder transitions

We now can explore how this kinetic form of RNA selection is influenced by changes in the control parameter. One reason that is necessary for us to do that is that the assembly time-scale of the MLD = 9, $N_p = 8$ spanning trees was in the range of $10^4$ time units. This is much too long, of the order of five days if one uses the earlier estimate that a time unit is about five seconds. Since assembly time kinetics depends exponentially on $E_0$, the assembly time can be reduced by reducing the energy scale $E_0$ but what happens with the selectivity if one reduces

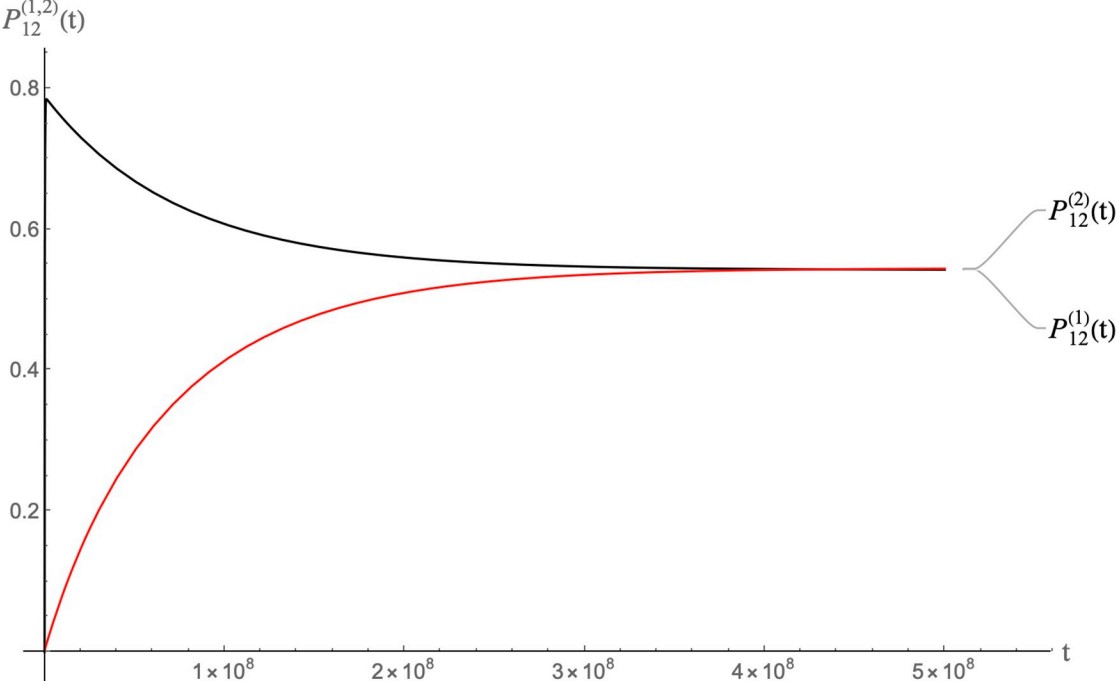

**Fig 10. Packaging competition between MLD = 9, $N_p = 8$ spanning trees and MLD = 19, $N_p = 2$ spanning trees with the same parameters as the previous figure on a time scale of $10^8$ units.**

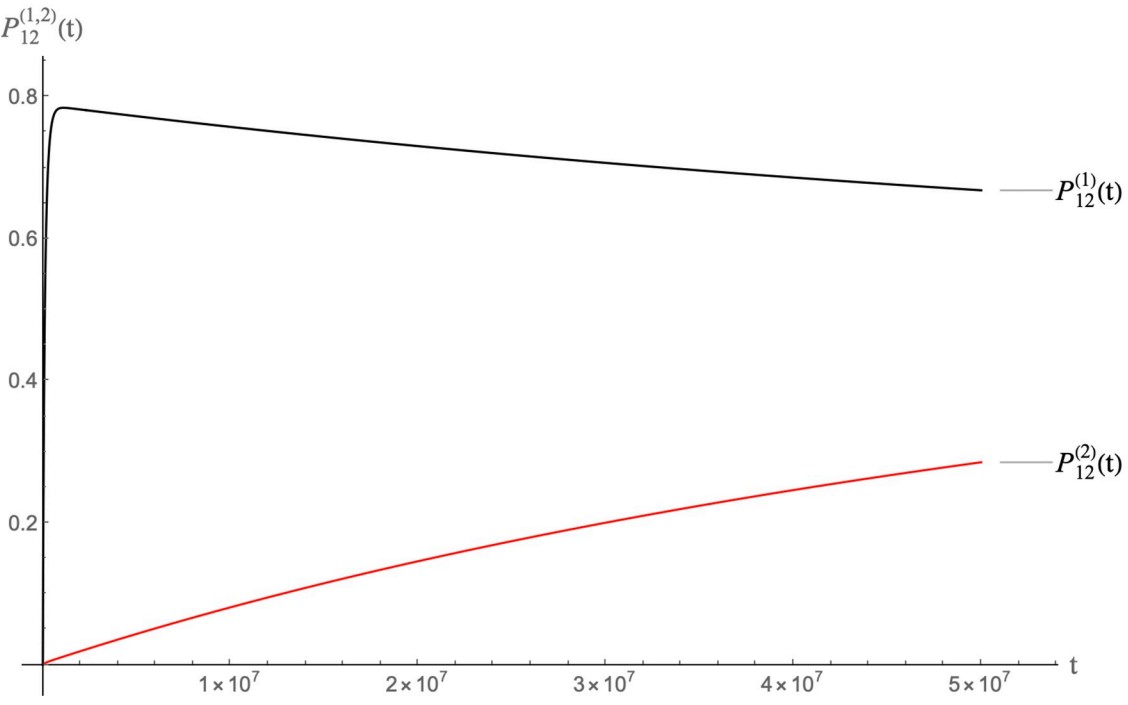

**Fig 11. Same as the previous figure but on a time scale of $10^7$ units.**

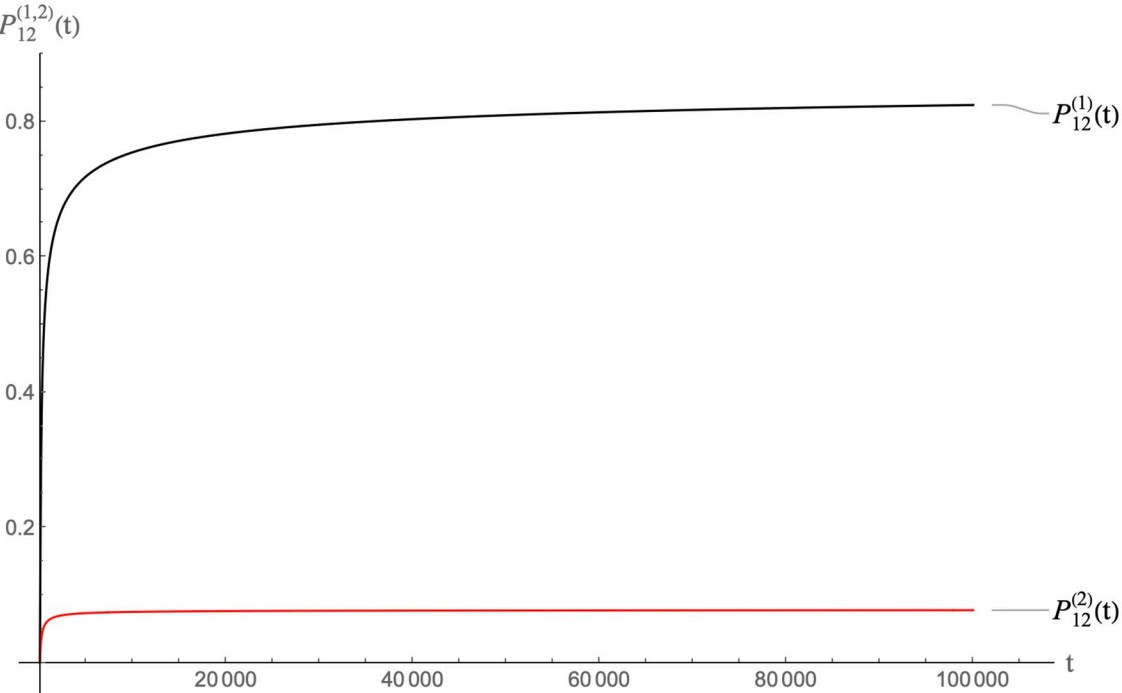

**Fig 12. Same as the previous figure but on a time scale of $10^4$ units.**

$E_0$? We will quantify the kinetic selectivity of a packaging competition between class (1) spanning trees that have a small MLD and a large wrapping number and class (2) spanning trees with large MLD and small wrapping number as

$$S(E_0) = \frac{P_{12}^{(1)}(t_{max}) - P_{12}^{(2)}(t_{max})}{P_{12}^{(1)}(t_{max})}. \tag{6}$$

Here, $t_{max}$ is the time at which class (1) spanning trees achieve their maximum packaging yield. If class (2) molecules outcompete class (1) then we set $S(E_0) = 0$. The dependence of the kinetic selectivity on the energy scale on $E_0$ is shown in Fig 13. The parameter values are the same as those of Fig 10. For $E_0$ less than about $1.2 k_b T$ there is no kinetic selectivity left while $S(E_0)$ approaches one for $E_0 \simeq 4.0$ or larger. The disappearance of selectivity for small $E_0$ is due to the fact that both the number of assembly pathways and the number of nodes for assembly intermediates of the MLD = 19, $N_p$ = 2 trees is two orders of magnitude larger than that of MLD = 9, $N_p$ = 8 spanning trees (see Fig 6). This means that for MLD = 19, $N_p$ = 2 trees the entropic component of the *free* energy activation barrier causes that barrier to be reduced by a factor of about four to five $k_b T$. Since the activation energy barrier is about $2 E_0$ higher for the MLD = 19, $N_p$ = 2 trees, we indeed should expect kinetic selectivity to become negligible for $E_0$ less than roughly $2 k_b T$, in agreement with Fig 13. The importance of entropy for smaller $E_0$ has another aspect. For $E_0$ less than about $1.5 k_b T$ there is a significant fraction of *incomplete* assemblies since that also increases the entropy of the system. In the language of statistical physics, the kinetic selectivity resembles the order parameter of an order-disorder phase-transition with the energy scale $E_0$ acting as the inverse of the effective temperature.

One encounters a somewhat similar transition for larger $E_0$ when one increases the affinity ratio $-\epsilon_1$ between specific pentamer/RNA affinity and pentamer/pentamer affinity. One might expect that that should improve selectivity but, in actuality, for $-\epsilon_1 = -1.1$ a variety of incomplete particles appear packaging MLD = 9, $N_p$ = 8 spanning trees. These incomplete particles are in coexistence with fully packaged particles containing MLD = 19, $N_p$ = 2 trees. The reason is shown in Fig 14. The minimum energy state at assembly equilibrium of MLD = 9, $N_p$ = 8 spanning trees are incomplete particles with ten pentamers while for MLD = 19, $N_p$ = 2 trees fully assembled particles are the minimum energy state. For $-\epsilon_1$ smaller than 1.0, the minimum energy state of MLD = 9, $N_p$ = 8 spanning trees are fully packaged particles. However metastable states with $n$ less than twelve start to appear roughly above $-\epsilon_1 = 0.6$. Metastable intermediate states are quite familiar from experimental studies of viral assembly [4, 62, 63] and from numerical simulations [13, 36–38]. Known as "kinetic traps", they retard assembly. In summary, It is clear that increasing $-\epsilon_1$ beyond about 0.5 also does not improve selectivity.

## Supersaturation

A different approach to reduce the assembly time scale is to increase the level of supersaturation. Recall in this context that assembly experiments under in-vitro conditions take place at relatively high levels of supersaturation. The supersaturation can be increased by increasing the total pentamer concentration $c_0$, which increases the chemical potential by $k_b T \ln c_0$. In Fig 15 the parameters are the same as in Fig 10 except that the pentamer concentration has been increased from $c_0 = 1$ to $c_0 = 4$. The results look encouraging. First, nearly all MLD = 9, $N_p$ = 8 spanning trees are packaged. Second, the assembly time scale is reduced to about $10^3$ time units, or about one hour. Increasing the supersaturation further reduces assembly time scales. However, the kinetic selectivity *also* has been reduced: the concentration of $N_P$ = 2, $MLD$ = 19 assemblies *also* rises much more rapidly than for $c_0 = 1$ under assembly equilibrium

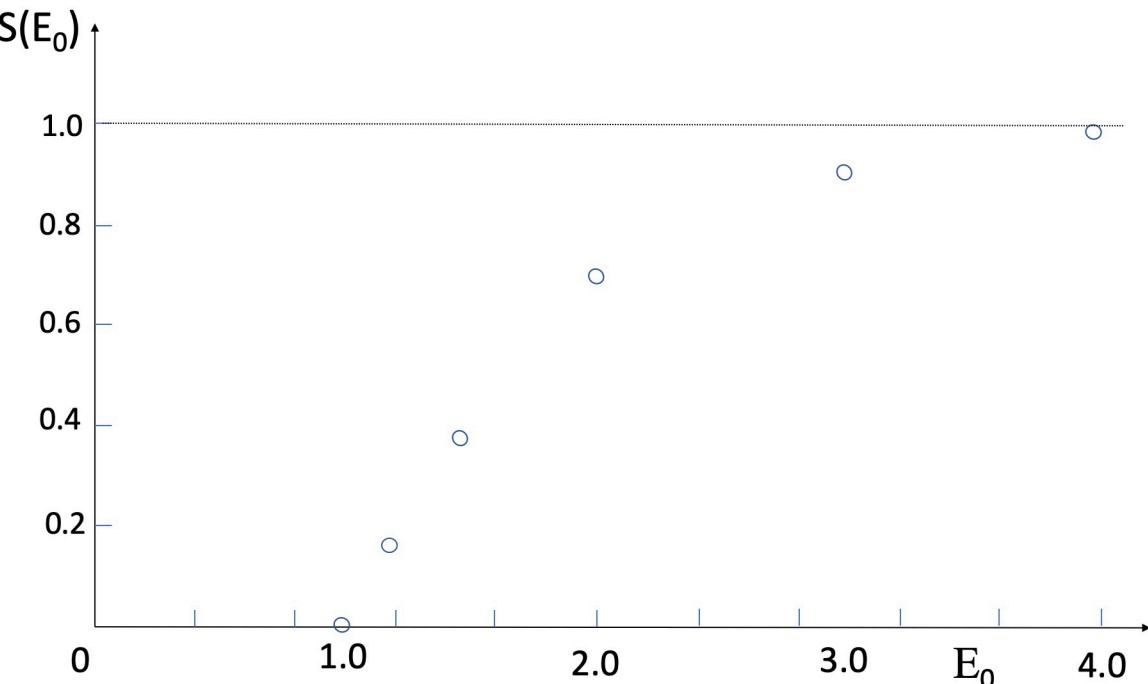

**Fig 13. Kinetic selectivity $S(E_0)$ as a function of the energy scale $E_0$ in units of $k_b T$ for packaging competition between MLD = 9, $N_p$ = 8 spanning trees and MLD = 19, $N_P$ = 2 spanning trees.** The other parameter values are the same as those of Fig 10.

conditions. The two classes of molecules have comparable packaging probabilities already after about $10^4$ time units.

Is it possible to maintain a high selectivity under conditions of supersaturation that persists over very long time scales? So far we kept the mixing ratio at $D$ = 0.5, meaning that the number of pentamers is double of what necessary to package all MLD = 9, $N_p$ = 8 and all $N_P$ = 2, $MLD$ = 19 spanning trees. What would happen if, under conditions of supersaturation, the

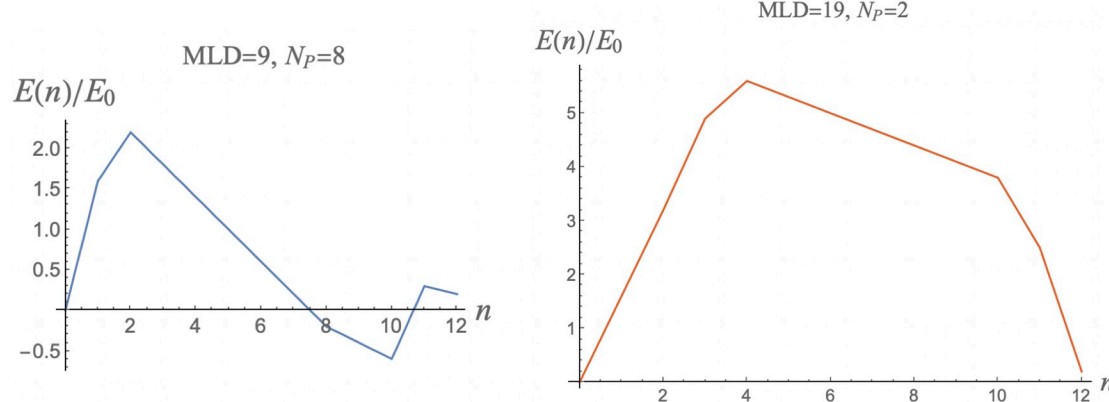

**Fig 14. Minimum-energy assembly profiles for $N_P$ = 8, $MLD$ = 9 spanning trees (left) and for $N_P$ = 2, $MLD$ = 19 spanning trees (right) for $-\epsilon_1 = -1.1$.**

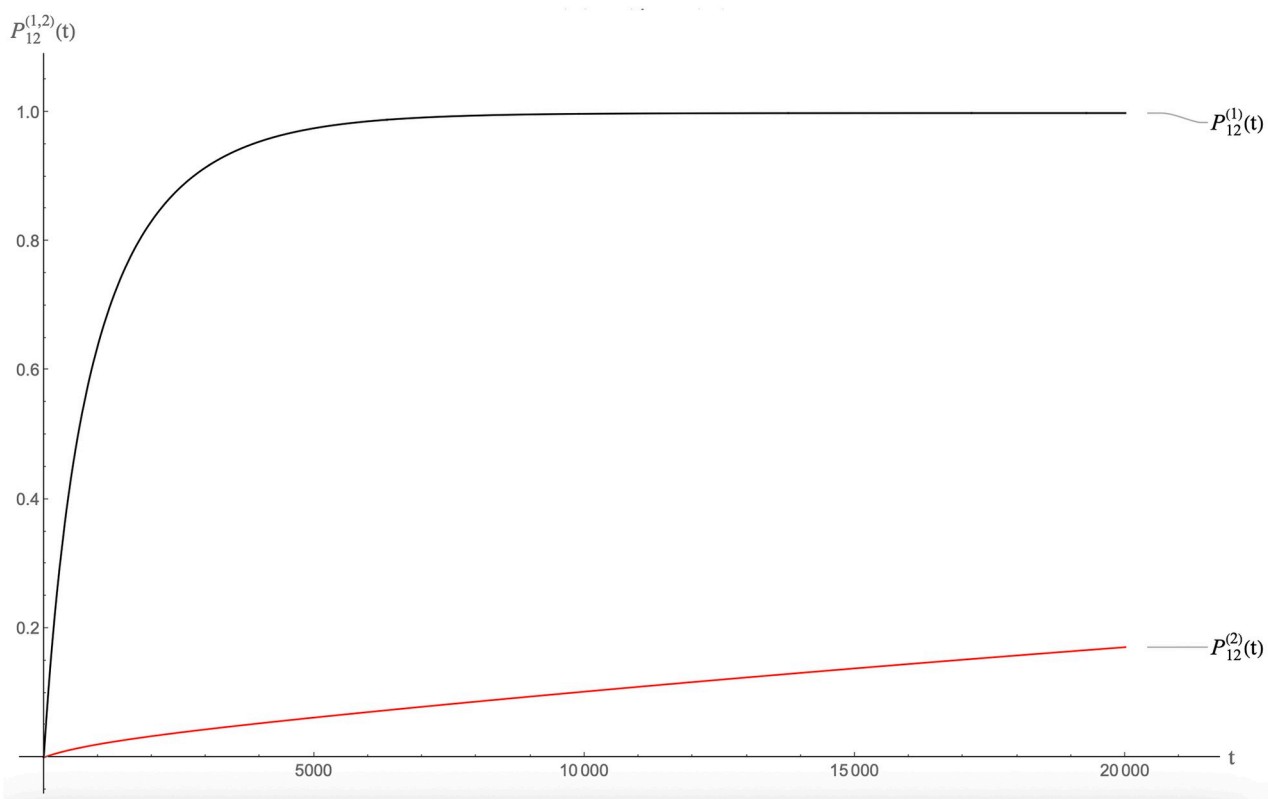

**Fig 15. Packaging competition for $c_0 = 4.0$.** The other parameters are the same as for Fig 10.

mixing ratio would be increased to $D = 2$? In that case, the early assembly of MLD = 9, $N_p = 8$ spanning trees might "drain" the solution of pentamers, which would delay the assembly of $N_P = 2$, $MLD = 19$ spanning trees. Fig 16 shows what happens. The kinetic selectivity is about 99 percent and it is maintained over more than $2 \times 10^5$ time units! It comes at a price however: the fraction of packaged target molecules is reduced to about 80 percent while the assembly time-scale has mildly increased. In general, by varying the overall pentamer concentration $c_0$ and the mixing ratio $D$ a wide range of requirements can be satisfied in terms of persistent kinetic selectivity, overall yield, and assembly time-scale.

## Parameter values

Are the energy and concentration parameters settings used in this article reasonable for in-vitro or in-vivo viral assembly experiments? The overall energy scale $E_0$ was defined as the binding affinity between two pentamers that share an edge for the case that the RNA molecule has only generic affinity (i.e. $\epsilon_1 = 0$). The assembly energy per capsid protein of empty capsids has been been measured under conditions of thermodynamic equilibrium [64]. Comparing such data to the model with $\epsilon_1 = 0$ gives $E_0 \simeq 4.73$ in units of $k_b T$ [50], close to the value $E_0 = 4$ used in the paper. The other important energy scale is the dimensionless parameter $-\epsilon_1$, which is the ratio between the protein/RNA and the protein-protein affinity. It can be estimated by comparing the capsid protein concentrations at assembly onset for empty capsids and for viri-ons. MS2 capsid proteins aggregate in RNA-free physiological solutions for concentrations above 2.0 mg/ml while in the presence of viral RNA (but not generic RNA) viral particles form

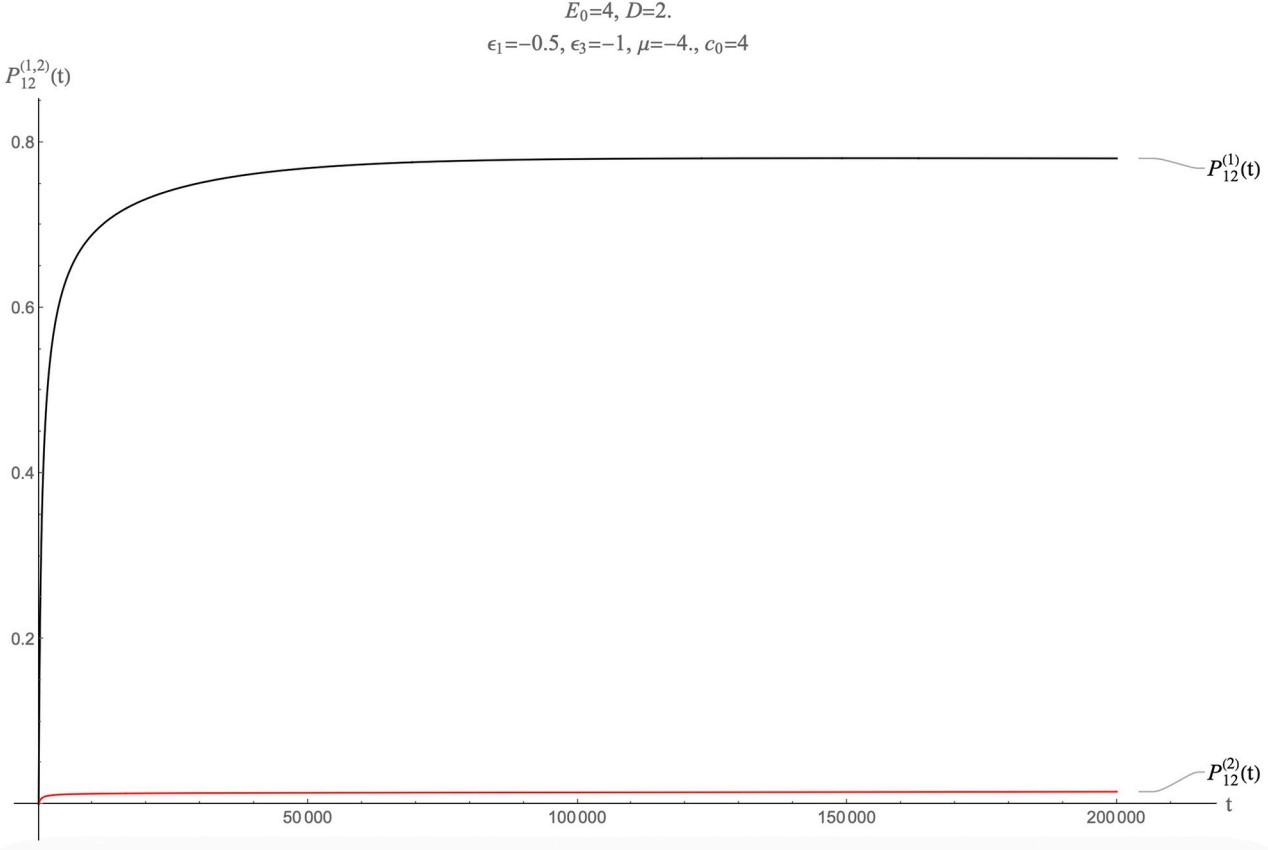

**Fig 16. Packaging competition for $c_0$ = 4.0 and $D$ = 2.** The other parameters are the same as for Fig 10.

already at 0.05 mg/ml [65]. The assembly free energy of a viral particle $E_0(-30 + 38\epsilon_1 - 12\mu_0)$ at the reference concentration was set to zero by definition, in which case the empty capsid assembly energy is $-30 - 12\mu_0$ is positive. An increase in the total pentamer concentration by a factor of 2.0/0.05 = 40 must be able to raise the chemical potential sufficiently so the assembly energy for an empty capsid becomes zero. An increase of the chemical potential equals $ln40$ in units of $k_bT$ or about 0.92 in units of $E_0 = 4k_BT$. This condition is satisfied if $-38\epsilon_1 = 12 \times 0.92$ or $-\epsilon_1 \simeq 0.29$. For convenience we used $\epsilon_1 = -0.5$ in the calculations.

## Conclusion

In conclusion, we have presented a statistical-mechanics model for the selection of viral ssRNA molecules triggered by packaging signals during the assembly of small RNA viruses. According to the model, there are two important time scales: the characteristic time scale for assembly and the characteristic time scale for thermal equilibration. In the model, RNA selectivity is a non-equilibrium, kinetic effect that disappears when the system approaches thermodynamic equilibrium. Particle *disassembly* under the action of thermal fluctuations is an essential step for thermal equilibration during packaging competition. Kinetic selection "works" as long as the time scale for spontaneous disassembly is prohibitively long compared to the measurement time. Kinetic selection is the result of the dependence of the height of the activation energy barrier on the RNA folding geometry, as encoded by the wrapping number

$N_p$ and the maximum ladder distance MLD. There is an order-disorder type transition as a function of the strength of the affinity between the capsid proteins where fully packaged particles are replaced by a polydisperse solution of incomplete particles. Near the transition, RNA molecules that optimally encode secondary structure and folding geometry—by having minimal MLD and maximal $N_p$—begin to outcompete other RNA molecules that have a larger number of assembly pathways. A similar order-disorder transition is encountered when the strength of the protein-RNA interaction is increased with respect to that of the protein-protein interactions.

The most striking result is the fact that kinetic selection performs better under increasing levels of supersaturation. There are in fact numerous examples in molecular biology where the fidelity of the read-out of a code is enhanced under non-equilibrium conditions, known as *kinetic proofreading* [66]. DNA replication is an important example [67, 68]. There are similarities between the kinetic selection under supersaturation discussed in this paper and Hopfield kinetic proof reading. Kinetic selection works when the formation of the encoded system (MLD = 9, $N_p$ = 8 spanning tree particles) is quasi-irreversible while attempts of forming particles with molecules that are not encoded (MLD = 19, $N_p$ = 2 spanning tree particles) lead to disassembly. This is the case because of the higher assembly activation energy. An unusual feature is that by tuning the mixing ratio to be close to the stoichiometric ratio for the early assembling encoded particles can "monopolize" the pentamers and thereby greatly retard the formation of improper particles.

An important question about the model concerns the relation between the kinetic selectivity discussed in this paper and selection in terms of the thermodynamic assembly free energy as discussed in the literature cited in the introduction. The relation between the two forms of selection lies in the distinction between the nucleation and elongation stages. An ssRNA molecule may well have packaging signals that produce an unusually low assembly energy barrier causing it to be selected during the nucleation stage. If, however, the molecular weight of the molecule is too high and/or if the solution radius of gyration is too large then the elongation process simply would not be able to complete. It is this elongation part of the assembly that was captured by the earlier studies of the thermodynamic assembly free energy and that was not included in the present study. Following ref. [45], we are assuming here that the selective effects of the packaging signals operate nearly exclusively during the nucleation stage and are weak during the energetically downhill elongation stage which is dominated by the non-specific interactions. The model presented in this paper can be generalized to investigate the competition between selection by nucleation and selection by elongation. The simplest step would be by letting the competing molecules have different reference chemical potentials $\mu_0$. This allows for the possibility that the two RNA condensates have a different molecular mass and/or a different overall MLDs (recall that the MLD in this paper only refers to the $\psi$ sequence). An additional refinement would be to allow $\mu_0$ to depend on $n$. As an assembly intermediate grows, an RNA molecule that is only partially condensed will be progressively confined by the developing capsid. This reduces the RNA conformational entropy, which reduces the free energy gain obtained when a pentamer is taken from the solution and added to an incomplete assembly. Finally, the loss of conformational entropy of individual RNA nucleotides as they are getting packaged can be included in the model by associating a fixed amount of entropy with every RNA segment that has not yet associated with a pentamer. This amounts to a renormalization of the $\epsilon_1$ and $\epsilon_2$ parameters.

A second concern with the model is that it is generally assumed that virions are in a state of full thermodynamic equilibrium. If that were truly the case then equilibrium thermodynamics rather than kinetics would *always* be the appropriate description mode. We would argue against the assumption of full thermodynamic equilibrium for virions. Reaching complete

thermodynamic equilibrium in a packaging competition experiment necessarily involves disassembly. The time-dependent assembly yield of early-assembly kinetically encoded molecules needs to decrease in order for thermal equilibrium to be established. Experimentally, the time scale for spontaneous disassembly of virions under the action of thermal fluctuations must be extremely large compared to observation times. We know this because under in-vivo conditions virions typically are released after assembly into environments with few or no capsid proteins. Under those conditions, the chemical potential of the capsid proteins of a virion would be large compared to that of capsid proteins free in solution, which would trigger disassembly. In actuality, spontaneous dissolution of virions in solutions without capsid proteins is not observed under physiological conditions. The time scale for spontaneous disassembly of virions must be large compared to both laboratory time scales and the characteristic time scales of the life cycle of a virus. Maturation processes [63, 69–77] probably play an important role in suppressing spontaneous disassembly process. Maturation stabilization of complete assemblies would further justify a focus on selection during nucleation, as opposed to selection by late-time disassembly/assembly competition necessary for thermalization.

A third concern is that the model presented in this paper is not a realistic representation of any particular virus. It was constructed, for mathematical convenience, by borrowing features of the dodecahedral Zlotnick model for empty capsids, the dodecahedral gRNA spatial distribution of the *Nodaviridae*, and the asymmetric reconstruction of MS2. The wrapping number concept as a geometric characteristic of the geometry of the RNA outer surface really is appropriate only for the special case of *Nodaviridae*. Different geometrical characteristics will have to be developed for other viruses. The MS2 virus is an interesting target since a detailed asymmetric reconstruction of the MS2 virion is available [44]. Since the packaging signals of MS2 gRNA associate directly with capsid proteins—rather than with capsomer edges—the spanning tree would have to have icosahedral rather than dodecahedral symmetry. Moreover, MS2 has (approximate) T = 3 capsid symmetry with 180 capsid proteins rather than the T = 1 structure with 60 proteins grouped in 12 pentamers that we assumed so that would have to be included as well. We hope to carry out such a study in the future.

A final concern about the model is the study by Tresset and collaborators of the assembly of the CCMV plant virus [78, 79]. They find that the so-called *en-masse* assembly scenario provides a good description [80]. In that scenario, a virion does not assemble on a protein-by-protein basis, as was assumed in the present paper. Instead, assembly starts with the formation of a disordered RNA-protein condensate that shrinks and then transits into an ordered virion. It is tempting to identify CCMV-type assembly (and the *en-masse* assembly scenario) with the entropy-dominated low selectivity assembly scenario that we encountered for lower $E_0$ and for larger values of the RNA/pentamer affinity. This also will have to be explored in the future.

How could the model (or one of its generalizations) be tested experimentally? It has been shown that large ssRNA molecules in solution with identical primary sequences adopt a range of secondary and tertiary structures [81]. In the presence of a sufficient concentration of positive polyvalent counter ions, large ssRNA molecules have been shown to condense [82]. For a solution of gRNA molecules, that should produce a variety of pre-condensed molecules with roughly the same size but with different surface structures. Asymmetric reconstruction of the packaged particles could then reveal which RNA structures were selected for.

## Supporting information

**S1 Text. Smoluchowski theory of bimolecular reactions.**
(PDF)

## Acknowledgments

We would like to thank Alexander Grosberg for suggesting the spanning tree, Ioulia Rouzina for suggesting selective nucleation as a central mechanism of viral assembly, Chuck Knobler and Reidun Twarock for reading a draft of the manuscript and Ioulia Rouzina, Orlando Guzman, Chuck Knobler, William Gelbart, Chen Lin, Zach Gvildys and William Vong for helpful discussions.

## Author Contributions

**Conceptualization:** Inbal Mizrahi, Robijn Bruinsma, Joseph Rudnick.

**Formal analysis:** Robijn Bruinsma, Joseph Rudnick.

**Investigation:** Inbal Mizrahi, Joseph Rudnick.

**Methodology:** Joseph Rudnick.

**Resources:** Robijn Bruinsma.

**Supervision:** Robijn Bruinsma, Joseph Rudnick.

**Validation:** Joseph Rudnick.

**Visualization:** Inbal Mizrahi.

**Writing – original draft:** Robijn Bruinsma, Joseph Rudnick.

**Writing – review & editing:** Inbal Mizrahi, Robijn Bruinsma, Joseph Rudnick.

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
