## [Decision Letter · Decision Letter 0]

6 Sep 2021

Dear Dr. Bruinsma,

Thank you very much for submitting your manuscript "Packaging Contests between Viral RNA Molecules." for consideration at PLOS Computational Biology.

As with all papers reviewed by the journal, your manuscript was reviewed by members of the editorial board and by several independent reviewers. In light of the reviews (below this email), we would like to invite the resubmission of a significantly-revised version that takes into account the reviewers' comments.

Though the work is promising, I cannot make any decision about publication until we have seen the revised manuscript and your response to the reviewers' comments. Your revised manuscript is also likely to be sent to reviewers for further evaluation.

Sincerely,

Samuel Coulbourn Flores, Ph.D.

Guest Editor

PLOS Computational Biology

Arne Elofsson

Deputy Editor

PLOS Computational Biology

Reviewer's Responses to Questions

**Comments to the Authors:**

Reviewer #1: The authors develop a kinetic model for the assembly of small RNA viruses based on the nucleation theory. As inside a host cell, there is a significant amount of non-viral RNA and other anionic polyelectrolytes, it is not clear how the capsid proteins select their native RNA. The main focus of the paper is on understanding how genomic RNAs outcompete other non-viral RNAs. The authors introduce the concept of the wrapping number, a geometrical measure of the ability of genome to bind capsid proteins. The authors discuss that the outcome of the competition can be explained in terms of the combined effect of the wrapping number and the maximum ladder distance (MLD), a topological measure of the degree of branching of the RNA secondary structure.

The paper is relatively well-written. I think that the subject of the paper is timely considering the current pandemic and the results are exciting. Thus I recommend the paper for the publication. However, there are several items that the authors need to clarify before publishing the paper.

1. The authors explain the assembly of capsid proteins around ssRNA based on the heterogenous nucleation. However, they discuss that “the spatial distribution of packaging signal is likely to play an important role.” If the process is based on the nucleation and growth, then why is the distribution of packaging signals important? After the capsid is nucleated in one of the packaging signals, does it still matter that there are other packaging signals if the process is based on the nucleation and growth? One might think about the process of multi-nucleation on different segments of the RNA, but it seems that the authors do not agree with the en mass assembly or multi nucleation growth. For the en mass mechanism and the multi-nucleation process see ACS Nano 14, 3170-3180 (2020).

2. The authors ignore the importance of the length of genome, which is often longer than mRNA. Recently, it was shown that the capsids assembled around shorter RNAs are unstable, see Small 2020, 16, 2004475. See also Physics Reports 847, 1-102 (2020) and many references therein.

3. If there are a few long branches in an RNA, it is not obvious if MLD is still a good “predictor” of RNA branchiness. Have you seen the work of van der Schoot and collaborators? For example, they show how the free energy or osmotic pressure due to the presence of RNA changes with the degree of RNA branching, see for example Phys. Rev. E 94, 022408, (2016). Since the base-pairing increases the stiffness of RNA segments, sometimes the linear polymers win over the branched ones to attract proteins (J. Phys.: Condens. Matter 30, 044002 (2018)). Also see The Journal of Physical Chemistry B, 119, 13991-14002 (2015) how branchedness and length contribute to the competition experiments.

4. Can the authors explain why they believe that the RNA selection mechanism is purely kinetic and disappears after the system has reached full thermodynamic equilibrium. Several papers show that the encapsidation of branched polymers is energetically more advantageous than a linear one, see Elife 2, e00632 (2013) in addition to the paper mentioned in item 3.

5. The authors find that the en-masse scenario is not a viable route for selective virion assembly. Have they seen the experiments of Tresset and collaborators, which show that for CCMV, en mass mechanism is a viable one (Nat. Commun. 9, 3071 (2018)), also see the simulations corresponding to his experiments in ACS Nano 14, 3170-3180 (2020).

6. The authors state, “Such “specific” interactions operate against a background of a generic electrostatic affinity between the negatively charged RNA nucleotides and positively charged residues of the capsid proteins.” Why does it work against? Doesn’t specific interaction facilitate the packaging, thus helping the electrostatic interactions?

7. There are a few redundant sentenceds and typos in the paper. I am sure that the authors will proofread it again.

Reviewer #2: Mizrahi et al. examine the selective packaging of RNAs into viral capsids. Their model assumes capsids are built from 12 pentameric subunits and the RNA interacts along edges of the pentamers. They then examine packaging competitions between different shaped RNAs which can be characterized by their maximum ladder distance.

The work in this manuscript is similar to that done by Dykeman and Twarock on the assembly competition between viral genomic RNAs and host cellular mRNAs that would occur in vivo, but contains a couple of key differences which make it new.

1) The authors model treats the interaction between RNA and capsid as occuring between pentameric edges. This type of model would be applicable to insect viruses in the Nodaviridae family which construct a dodecahedral cage of ds-RNA. An assembly model looking at this type of interaction (along with a combinatorics enumeration of the RNA configurations) has not been constructed yet.

2) The model attempts to account for some of the effects of the geometry of the RNA secondary structure and its effects on RNA assembly efficiency which is lacking in previous models (e.g. Dykeman et al.).

Some comments on the science.

MAJOR

1) Enumeration/Combinatorics of the spanning trees. There is limited discussion on the overall numbers of spanning trees and the computational method how they were counted. The manuscript also leaves out an important discussion on the number of ways to embed a spanning tree. For example. The linear spanning tree case corresponds to the previous treatment of RNA paths in Dykeman et al. For this one case, there are ~ 64 ways to embed the linear spanning tree to obtain a Hamiltonian path. Similarly, there will be an equivalent treatment for the other spanning tree shapes which will each have a potentially different number of ways to embed the spanning tree into the dodecahedron protein shell resulting in a Hamiltonian path. The Authors use the minimum energy embedding, where the pentamers are brought onto the RNA in an order which minimizes the energy at each step. This seems reasonable, but could have some issues if there is more then one path which does this. This would give an entropic term that needs to be accounted for where spanning trees with more path options that have the same energetic pathway are favored over others with less options. For example, in Fig 1 my guess (unless you tell me I am mistaken) is that there are two orientations of the path which should have the same kinetic profile in your model. The one shown, and the other where the order of the four edges on one of the two blue pentamers is clockwise instead of anti-clockwise.

Paths are fairly easy to construct on the dodecoahedron, so I don't think it should be to hard to enumerate the embedding options (only 64 for the linear case with less options I would guess for more complicated trees). But this is dependent on the Authors spanning tree enumeration procedure which they should discuss.

2) When constructing minimum energy paths, the manuscript states that (line 168) for more then one choice of lowest energy configuration, an arbitrary choice is made. However, this choice could impact on the choices further down the path, resulting in lower drops in free energy further down the path which may be kinetically less favorable. It is probably not too much of an issue with the doedec since it is small, has short assembly with limited options, but some discussion about this issue would be nice. Potential fixes would be a breadth first search of the assembly paths, keeping the n lowest energy at each pentamer addition. This is also a simple check to see if there is more then one embedding which has the same energy at each step (point 1 above).

3) The choice of energy parameters needs more justification. In Zlotnick (1994) the protein-protein interaction energy was chosen such that fully assembled empty particles were obtained at thermodynamic equilibrium with almost no free capsid. The critical threshold value for this was around -2.8 kCal/Mol (which depends on concentration of total protein - your c0). I would have thought that it would be ideal to be somewhere around this value since empty particles for most viruses form readily at appropriate concentration in vitro, but it is not clear where your parameters lie. (see point 4 in MINOR) Some more clarification here would be appreciated.

MINOR

1) The model for RNA interactions with the capsid has a Hamiltonian path on a dodecahedron and a dodecahedron for the protein shell. In Dykeman et al. the capsid was a dodecahedron and the Hamiltonian path was on an icosahedron. This was due to RNA-CP interactions being modeled as occurring at the centers of the capsids. Here both the protein cage and Hamiltonain path cage are dodehaedron, with RNA interactions on the edges. There is nothing wrong with this per se, but I would clarify that this models the type of situation in Nodaviridae. Some discussion of how the model could be used in viruses such as MS2 would also be useful.

2) A brief discussion on why (mathematically) every spanning tree for the dodecahedron must have 20 nodes and 19 links would be useful.

3) Similarly, the wrapping number and its connection with the number of pentamer edges occupied by RNA should also be discussed more. There is important nuisances between the linear case and the branched case, i.e. in the linear case, pentamers must have either 2/3/4 edges occupied by RNA, for any embedding (as the Hamiltonian path requirement enforces this when there are only 2 end nodes as in the linear case) where as for the branched case, there could be embeddings which have pentamers with 0/1/2/3/4 edges occupied (due to > 2 end nodes). An end node is a vertex on the spanning tree with only one link emanating from it.

4) The values for the energy parameters (in particular e1,e2,e3,e4) are somewhat convoluted into dependencies on D,c0 etc.. I would suggest also computing the values for e1-e4 for your examples and state concentrations of co in equivalent values of uM, as this will allow for comparisons with expected concentration in cells and in vitro experiments.

5) For the model, is there an intermediate step (where RNA-CP contact is formed followed by CP-CP contact) or do both RNA-cp and CP-CP contacts occur in one step?

6) Some more discussion on the nucleation energies and a comparison between the different spanning trees would be interesting.

7) There was no discussion (as promised) about ways to incorporate RNA conformation entropy into the model in the conclusions. Some discussion on how RNA secondary structure can be mapped onto your spanning trees (e.g. where are single stranded and double stranded areas on the tree, where are LD interactions and local hairpins). I think your model is currently applicable to Nodaviridae, but I would be interested in a brief discussion of how it could be applied to MS2.

**Have the authors made all data and (if applicable) computational code underlying the findings in their manuscript fully available?**

Reviewer #1: **No: **The computational code is not available.

Reviewer #2: Yes

PLOS authors have the option to publish the peer review history of their article (what does this mean?). If published, this will include your full peer review and any attached files.

Reviewer #1: No

Reviewer #2: **Yes: **Eric C. Dykeman
---

## [Decision Letter · Decision Letter 1]

9 Feb 2022

Dear Dr. Bruinsma,

We are pleased to inform you that your manuscript 'Packaging Contests between Viral RNA Molecules and Kinetic Selectivity.' has been provisionally accepted for publication in PLOS Computational Biology.

I think the writing is very didactic and the idea is original and interesting. Best of luck with the rest of the process.

Best regards,

Samuel Coulbourn Flores, Ph.D.

Guest Editor

PLOS Computational Biology

Arne Elofsson

Deputy Editor

PLOS Computational Biology

Dear Authors,

I like the clear and didactic explanation, though it is a bit outside my usual field of work.

The referees appear to be largely positive to the work, I will ask the editorial staff to send your revision 1 to them to see if they have any further comments.

For my part I have only a very minor pedantic comment -- on line 554 please fix "adopt will adopt".

With kind regards,

Sam

Reviewer's Responses to Questions

**Comments to the Authors:**

Reviewer #1: The authors have modified the manuscript based on my recommendation and addressed all my concerns. I thus recommend for publication now.

Reviewer #2: The manuscript has substantially improved and the Authors have thoughtfully replied to my concerns in the revision. I am happy with the revisions and I believe that it can now be accepted.

**Have the authors made all data and (if applicable) computational code underlying the findings in their manuscript fully available?**

Reviewer #1: Yes

Reviewer #2: Yes

PLOS authors have the option to publish the peer review history of their article (what does this mean?). If published, this will include your full peer review and any attached files.

Reviewer #1: No

Reviewer #2: **Yes: **E C Dykeman

---

## [Editor Report · Acceptance letter]

28 Mar 2022

PCOMPBIOL-D-21-01496R1 

Packaging Contests between Viral RNA Molecules and Kinetic Selectivity.

Dear Dr Bruinsma,

I am pleased to inform you that your manuscript has been formally accepted for publication in PLOS Computational Biology. Your manuscript is now with our production department and you will be notified of the publication date in due course.

With kind regards,

Agnes Pap
